# The Growing Importance of Tuberculosis Preventive Therapy and How Research and Innovation Can Enhance Its Implementation on the Ground

**DOI:** 10.3390/tropicalmed5020061

**Published:** 2020-04-16

**Authors:** Anthony D. Harries, Ajay M.V. Kumar, Srinath Satyanarayana, Pruthu Thekkur, Yan Lin, Riitta A. Dlodlo, Mohammed Khogali, Rony Zachariah

**Affiliations:** 1International Union Against Tuberculosis and Lung Disease, 68 Boulevard Saint Michel, 75006 Paris, France; akumar@theunion.org (A.M.V.K.); SSrinath@theunion.org (S.S.); Pruthu.TK@theunion.org (P.T.); ylin@theunion.org (Y.L.); rdlodlo@theunion.org (R.A.D.); 2London School of Hygiene and Tropical Medicine, Keppel Street, London WC1E 7HT, UK; 3International Union Against Tuberculosis and Lung Disease, South-East Asia Office, C-6 Qutub Institutional Area, New Delhi 110016, India; 4Yenepoya Medical College, Yenepoya (Deemed to be University), University Road, Deralakatte, Mangalore 575018, India; 5International Union Against Tuberculosis and Lung Disease, No.1 Xindong Road, Beijing 100600, China; 6Special Programme for Research and Training in Tropical Disease (TDR), World Health Organization, Avenue Appia 20, 1211 Geneva 27, Switzerland; khogalim@who.int (M.K.); zachariahr@who.int (R.Z.)

**Keywords:** tuberculosis, post-tuberculosis morbidity and mortality, TB preventive therapy, latent TB infection, Asia Pacific, rifapentine-isoniazid

## Abstract

Ending the tuberculosis (TB) epidemic by 2030 requires two key actions: rapid diagnosis and effective treatment of active TB and identification and treatment of latent TB infection to prevent progression to active disease. We introduce this perspective by documenting the growing importance of TB preventive therapy on the international agenda coupled with global data showing poor implementation of preventive activities in programmatic settings. We follow this with two principal objectives. The first is to examine implementation challenges around diagnosis and treatment of active TB. Within this, we include recent evidence about the continued morbidity and heightened mortality that persists after TB treatment is successfully completed, thus elevating the importance of TB preventive therapy. The second objective is to outline how current TB preventive therapy activities have been shaped and are managed and propose how these can be improved through research and innovation. This includes expanding and giving higher priority to certain high-risk groups including those with fibrotic lung lesions on chest X-ray, showcasing the need to develop and deploy new biomarkers to more accurately predict risk of disease and making shorter treatment regimens, especially with rifapentine-isoniazid, more user-friendly and widely available. Ending the TB epidemic requires not only cure of the disease but preventing it before it even begins.

## 1. Introduction

The international community has pledged to end the tuberculosis (TB) epidemic by 2030. The principal targets for meeting this ambitious goal include a 90% reduction in TB deaths and an 80% reduction in TB incidence rate by 2030 compared with 2015 [1]. On the ground, there are two key actions that need to take place if this goal is to be realised. First, every individual who develops active TB (estimated at about 10 million each year) must be rapidly diagnosed and effectively treated. This benefits not only the individual with active TB but also reduces the risk of further transmission of *Mycobacterium tuberculosis* (*MTB*) within the family, other close contacts and the community. Second, of the 1.7 billion individuals estimated globally to have latent infection with *MTB* [2], those at risk of progressing to active disease need to be identified and treated. 

While the prevention of TB makes intuitive public health sense, this has been a relatively neglected component of TB control efforts. However, recent years have begun to see a change. The World Health Organization’s (WHO) End TB Strategy, adopted by the World Health Assembly in 2014, emphasises preventive therapy in persons at high risk of TB under its first pillar of patient-centred care and prevention [1]. During the United Nations High Level Meeting (UNHLM) on the fight against TB in September 2018, world leaders committed to provide preventive therapy to at least 30 million people between 2018 and 2022. This included 4 million children aged <5 years, 20 million other household contacts of people diagnosed and treated for TB and 6 million people living with HIV (PLHIV) [3]. This translates to giving TB preventive therapy globally to 800,000 children aged <5 years per year, 4 million other household contacts per year and 1.2 million PLHIV per year. 

In 2018, the proportions of children aged <5 years (who were household contacts of people with bacteriologically confirmed TB) and PLHIV (newly enrolled in care) who were placed on TB preventive therapy were well below target at the global level and in the WHO South-East Asia and Western Pacific Regions (which comprise the Asia Pacific) (Table 1) [4]. Moreover, at the global level, only 79,000 other household contacts aged ≥ 5 years (2% of annual target) were given TB preventive therapy. Clearly a lot more work is needed to raise performance and get anywhere near the UNHLM preventive therapy targets. 

There are two main objectives of this perspective. The first is to outline the implementation challenges around the diagnosis and treatment of TB. This includes highlighting the morbidity, disability and heightened mortality that persist after successful treatment completion, in order for readers to appreciate the importance of TB preventive therapy in the overall context of ending the TB epidemic and ensuring healthy lives. The second objective is to describe how current TB preventive therapy activities have been shaped and are managed. We further assess some of the implementation challenges of TB preventive therapy and propose how activities can be improved and enhanced through research and innovation. 

## 2. Challenges with the Diagnosis and Treatment of Active TB

### 2.1. Screening and Diagnosis

There are two components to the efficient screening and diagnosis of TB. People with symptoms suggestive of TB must recognise that they are ill and seek appropriate and timely care, and the health services must respond quickly and efficiently. Unfortunately, symptomatic individuals can take on average 3 months or more to present to the health sector and the health system can take a further 1 month to diagnose TB [5,6]. Substantial delays from first symptoms to diagnosis are therefore often the norm. Large numbers of symptomatic individuals presenting to health facilities may also fail to be properly screened or investigated with appropriate microbiological or radiological tests and the diagnosis of TB therefore can be completely missed [7,8]. These problems of so-called “passive case finding” have plagued TB Control Programmes for decades and programmes frequently overlook the proportion of patients who suffer the fate of pre-diagnostic loss to follow-up. 

Active case finding, in which people who do not actively seek health care are proactively sought out and screened for TB regardless of symptoms, has the potential to overcome some of these limitations. Some pilot projects have shown that community-wide screening for active TB can reduce TB prevalence and TB infection, as shown recently in Vietnam [9]. However, an evaluation of several active case finding projects in 16 countries showed no impact on national case notifications mainly because the projects had not been taken to scale due to costs and shortages of human resources [10]. Other active case finding projects have found low detection yields of TB [11], and at the programmatic level are beset with challenges of inadequate health care worker training, staff shortages, community distrust, illiteracy and lack of community awareness about TB [12]. Much more investment and commitment are needed to take active case finding to scale and overcome multiple implementation challenges. 

Finally, an overview of 21 national TB prevalence surveys in Asia between 1990 and 2012 showed that between 40% and 79% of patients with bacteriologically-positive TB did not report TB symptoms and were only detected due to chest X-ray screening of all survey participants [13]. Such high proportions of asymptomatic TB patients present important challenges for standard screening and diagnostic algorithms and bring up the question about whether community use of chest X-ray should be expanded in the routine setting.

### 2.2. Initiating Anti-TB Treatment

While it should be straightforward to initiate treatment in all those diagnosed with TB, in practice this is far more difficult than it seems. In low- and middle-income countries in Africa, Asia and the Western Pacific between 4% and 38% of patients with laboratory-detected sputum smear-positive or culture-positive TB fail to start treatment [14]. This pre-treatment loss to follow-up appears to be no better with the use of rapid molecular technology [15,16,17]. For those who do get treated, turn-around times between confirmed diagnosis and treatment initiation can also be lengthy, compromising individual care and increasing the risk of *MTB* transmission within families and communities. 

### 2.3. Providing Effective Treatment

TB is a treatable and curable infectious disease. The duration of treatment for drug-susceptible TB is 6 months, with previous attempts to shorten this to 4 months with third generation fluoroquinolones and/or rifapentine being unsuccessful [18,19,20]. In contrast, there has been relatively good progress in shortening the treatment of multidrug-resistant TB (MDR-TB, resistant to isoniazid and rifampicin) from 24 months down to 9 or 12 months [21,22]. Trials are in progress to assess whether even shorter regimens of 6 months are effective and whether fully oral treatment with new or repurposed second-line drugs can be used [23]. 

Despite this progress with TB treatment regimens, treatment outcomes at the global level and in the Asia Pacific remain substandard (Table 2). Globally, treatment success for new and previously treated TB patients is below 85%, and this is even lower for patients with HIV-positive TB and drug-resistant TB [4]. In patients with new/previously treated TB, death, lost to follow-up and not evaluated are the main adverse outcomes while treatment failure, death and lost to follow-up are the main adverse outcomes in drug-resistant TB. Many of these outcomes are potentially correctable. For example, late detection of HIV-associated TB and delays in starting antiretroviral therapy (ART) and TB treatment account for most HIV-related TB deaths [4]. On the other hand, enabling local staff to make sense of their TB data has been associated with reduced losses to follow-up and better treatment success [24]. 

### 2.4. Taking Account of Post-Tuberculosis Morbidity and Mortality

It is widely thought that the outcome of “treatment success” signifies the end of TB treatment and a return to active healthy life. Indeed, most TB disease burden analyses, using disability-adjusted life-years (DALYs), assume that survivors return to full health post-tuberculosis [25]. However, a growing body of evidence shows that this is far from the truth. 

Many individuals who successfully complete TB treatment continue to be burdened with chronic pulmonary impairment from obstructive and restrictive lung disease that either pre-existed or developed as a result of TB [26]. Other morbidities such as permanent hearing loss from second-line injectable anti-TB drugs [27] and mental health disorders [28,29] compound these serious and long-term post-TB sequelae. Not surprisingly, a recent systematic review and meta-analysis found that all-cause mortality was nearly three times higher in individuals post-tuberculosis compared with age- and sex-matched controls [30]. A study using conservative estimates of post-TB morbidity and mortality from chronic obstructive lung disease only showed that the burden of TB in India would increase by 6.1 million DALYs—a 54% increase on the current estimates that assume a full return to health at the end of anti-TB treatment [31]. This burden would increase further if other post-TB conditions such as restrictive lung disease or permanent hearing loss from second-line injectable drugs were included. 

### 2.5. Implications of Shortfalls in Diagnosis, Treatment and Cure of TB

In summary, in 2018 only 7 million out of an estimated 10 million new patients with TB were notified. The rest, referred to as the ‘missing millions’, were not notified due to underdiagnosis and/or underreporting of detected patients [4]. Of the 7 million notified TB patients, about 5.8 million successfully completed treatment [4]. Good quality targeted operational research should lead to better programmatic implementation of proven interventions around diagnosis and treatment and therefore potentially better outcomes [32]. Prevention of TB, when coupled with improved diagnosis and treatment, would have an added and synergistic effect on TB control. With a considerable number of patients also continuing to experience significant morbidity and heightened mortality in the post-TB period, preventive therapy becomes even more important due to the greater DALYs averted per TB episode prevented. 

## 3. Current Status and Management of TB Preventive Therapy

The WHO provides guidelines for the programmatic management of latent TB infection (LTBI) and TB preventive therapy. The most recent published version was released in 2018 [33] and forthcoming changes are due to appear later in 2020 [34]. The key components of TB preventive therapy are briefly outlined below. 

### 3.1. Identifying High Risk Groups for TB Preventive Therapy 

Among those with LTBI, there are various populations at much higher risk of TB compared with the general population [35]. These high-risk groups need to be identified for testing and/or treatment of LTBI providing active TB is ruled out. The two highest priority groups are PLHIV and household contacts of people with bacteriologically confirmed TB. For PLHIV, while it is not necessary to systematically screen for LTBI because benefits of preventive therapy for all affected people outweigh any risks, those with positive results to tuberculin skin testing (TST) and interferon-gamma release assays (IGRAs) appear to benefit more from TB preventive therapy [33]. Similarly, for household contacts there is a higher ratio of benefit to risk from preventive therapy, and systematic screening for LTBI is only recommended for HIV-negative adults, adolescents and children ≥ 5 years if they are from a low TB incidence country (<100 patients per 100,000 population). 

Other HIV-negative high-risk groups are listed in Table 3. For those in Categories 1–3, systematic testing for LTBI is recommended if TB preventive therapy is to be considered, and those who test positive can be offered treatment. For those in Category 4, systematic testing for LTBI is generally not recommended but treatment can be offered on an individual basis if testing is done and is positive. 

### 3.2. Diagnosis of LTBI

There are no perfect ways of testing for and diagnosing LTBI. The two main tests in use (TST and IGRAs) both measure the immunological response to *MTB* antigens. TST is inexpensive but logistically challenging for patients and health care staff, different results may be obtained depending on which purified protein derivative products are used [37] and there continue to be widespread shortages of quality-assured tuberculin in many low- and middle-income countries [38]. IGRAs are more specific than TST and only one clinic visit is required by the patient for a venous blood sample. However, they are costly, there is a need for specialised laboratory equipment and there are issues related to reproducibility of results. The validity of both tests has recently been questioned with evidence to suggest that only 10% of persons showing immunoreactivity to TST or IGRAs harbour viable *MTB* organisms capable of causing disease [39]. 

### 3.3. TB Preventive Therapy Regimens

For many years, isoniazid preventive therapy (IPT), self-administered for 6–12 months, has been the mainstay of treatment for HIV-infected and HIV-uninfected persons and household contacts. Six-months IPT is highly effective in high-risk groups, it adds to the already considerable TB preventive benefits of ART in PLHIV and is recommended by WHO [33,36]. Amongst PLHIV in high TB exposure areas, for example in Southern Africa, continuous IPT for 36 months or longer provides more durable and robust TB prevention and is therefore recommended [40]. The likely reason for this effect is that isoniazid treats existing LTBI and also prevents new *MTB* infections from taking hold and progressing to active TB. 

Alternative and shorter regimens for treating LTBI are available (Table 4). In clinical trials these regimens appear to have the same efficacy as IPT but have higher rates of completion, improved medication adherence and better safety profiles [33,36]. The 3- or 4-month regimen of daily rifampicin and isoniazid (RH) is popular with children because of child friendly, dispersible, paediatric fixed-dose formulations. The 3- or 4-month regimen of rifampicin alone is not so widely used because of possible irrational use of the drug (which has broad spectrum antibacterial activity) to treat many non-TB conditions which could lead to widespread rifampicin resistance [36]. 

The 3-month weekly rifapentine and isoniazid (3HP) regimen, 12 doses in total, is rapidly becoming an attractive option and there is growing experience with this regimen from clinical studies and implementation in the field [33,41]. A recently completed trial of 4-weeks daily rifapentine and isoniazid (1HP) in adults or adolescents with HIV showed non-inferiority in preventing TB compared with 9-months IPT along with significantly better rates of treatment completion [42]. One of the important concerns with rifapentine has been the high cost. However, in 2018 the Global Drug Facility and the pharmaceutical company that manufactures the drug, Sanofi, reached agreement to offer rifapentine for USD$45 per patient course of 3HP, and activist pressure is being applied to produce generic patient-friendly formulations for as little as USD$10–USD$15 per patient course [43]. 

For household contacts of patients with MDR-TB or XDR-TB (MDR-TB with added resistance to fluoroquinolones and second-line injectable agents) there are only observational studies to guide recommendations [44,45,46,47]. The drugs used for MDR-TB contacts have been mainly fluoroquinolones with or without other drugs such as ethambutol or ethionamide. As about a quarter of the TB patients amongst households exposed to MDR-TB do not have drug-resistant disease [48], TB preventive treatment has to be individualised with selected drugs based on the drug susceptibility profile of the index patient. For XDR-TB contacts there is no specific guidance and currently close observation and follow-up is advised. 

## 4. Research and Innovation to Improve Delivery and Uptake of TB Preventive Therapy

### 4.1. Expanding the High-Risk Groups for TB Preventive Therapy

Currently, it is recommended that PLHIV newly enrolled in care and treatment receive TB preventive therapy [33]. The number of PLHIV newly enrolled in care is considered as the denominator against which WHO reports the percentage receiving TB preventive therapy on an annual basis [4]. However, giving TB preventive therapy to all PLHIV, regardless of how long they have been on ART, is likely to be beneficial and should be considered. Laboratory studies have shown that long-term recovery of TB specific immune function is incomplete on ART [49], and clinical studies have shown that length of time on therapy and ART-induced immune recovery still do not fully protect against TB in high exposure environments [50]. 

Household contacts of patients with bacteriologically confirmed TB are high priority for TB preventive therapy. There are several research questions that need to be answered around the index patient and the household contacts if practice is to be refined and improved (Table 5). The accepted definition of a household contact is “a person who shared the same enclosed living space for one or more nights or for frequent or extended periods during the day with the index case during the 3 months before commencement of the current treatment episode” [51]. This is an arbitrary definition and in the local context it needs to be tested out through operational research and adapted as necessary. For example, the amount of exposure to *MTB* will vary from sharing the same bed to living somewhere else within the same household complex and the actual duration of infectiousness may be longer or shorter than 3 months. 

Currently, only patients with end-stage renal disease are recommended for systematic testing and treatment of LTBI [33]. However, a well-conducted cohort study in Taiwan showed that there is an increased risk of TB in early stage chronic kidney disease (CKD) [52], and it has been suggested that TB prevention efforts be targeted to all people with this condition. This recommendation needs further study because over the last 25 years, the global all-age prevalence of CKD has increased by 29% with nearly 700 million patients of all-stage CKD recorded in 2017 [53]. 

In high TB burden countries, prisoners, people who inject drugs and health care workers are all at high risk of TB [35,36]. Operational research should be conducted to assess whether in the local context and based on available resources it is cost-effective to systematically enrol such groups into TB preventive therapy services. Finding ways to ensure adherence in prisoners and people who inject drugs will need operational research. 

Currently, WHO does not recommend that persons with diabetes mellitus (DM) be systematically screened and treated for LTBI [33]. This needs to be revisited especially in the Asia Pacific Region. Persons with DM have an overall three-times higher risk of TB compared with the general population [54,55]. A systematic review and meta-analysis estimated that the pooled prevalence of DM amongst TB patients between 1986 and 2017 was 15%, with the Asia Pacific in particular having a higher prevalence than other regions at 19% [56]. In Indonesia, TB incidence was found to be significantly higher in persons with DM with established LTBI (1.7 per 100 person-years) compared with those without LTBI (0.5 per 100 person-years) [57]. In Singapore, TB incidence was higher in persons with DM compared with the normal population and increased significantly in persons with DM as their body mass index dropped, being highest in those who were underweight [58]. Further research is needed to determine whether persons with DM should be targeted for systematic LTBI testing and TB preventive therapy. In this regard, a prospective randomised controlled study is approved and about to start in Tanzania and Uganda using 3HP (European Union–EDTCP2 programme and grant number RIA2018CO-2514-PROTID). If TB preventive therapy is found to be cost-effective and taken up by WHO, this would considerably expand the pool of people potentially eligible for LTBI testing and TB preventive therapy. In 2019, there were 463 million people living with DM (54% living in the Asia Pacific) and this is predicted to rise to 578 million by 2030 [59]. 

Finally, there is no mention in the WHO Guidelines about what to do with HIV-uninfected persons who have fibrotic lung lesions on chest X-ray. A trial in Eastern Europe 40 years ago found a high incidence of TB in this population group with TB preventive therapy using isoniazid significantly reducing this risk [60]. In this regard, the expanded use of chest X-ray should be further considered, and those with fibrotic lung lesions consistent with inactive TB could be assessed for LTBI testing and treatment. 

### 4.2. Better Tests for LTBI

There is an urgent need to develop and then deploy sensitive and specific biomarkers that can distinguish infection with *MTB* from immunological memory of past infection (which is essentially what TST and IGRAs do) and predict who will progress from LTBI to active TB disease. It is becoming clear that LTBI is not a single entity but rather represents a broad spectrum of asymptomatic TB infection where different degrees of inflammation, bacterial replication and host immunity determine whether disease will develop or not [61]. An exciting development in this direction has been the use of a whole blood transcriptomic messenger RNA expression signature that in Cape Town, South Africa, predicted progression from LTBI to active TB disease with 66% sensitivity and 81% specificity [62]. Further research is continuing in this direction [61], but currently there are no clinically useful or affordable tests for use in the field. 

### 4.3. Ruling Out Active TB

A “sine qua non” of TB preventive therapy is ensuring that no person with active TB starts mono- or dual therapy. Screening adults and children for suggestive symptoms of TB is recommended by WHO [33]. Those with symptoms are investigated and if TB is not diagnosed, TB preventive therapy can be considered. While molecular technology, particularly with Xpert MTB/RIF or Xpert MTB/RIF Ultra, has greatly improved the sensitivity and specificity of diagnosing active TB [63], diagnostic certainty cannot be guaranteed. For this reason, symptomatic persons are often not offered TB preventive therapy. 

The big question is whether absence of symptoms in adults or children is sufficient for ruling out active disease or whether chest X-ray should also be performed. The systematic use of chest X-ray is not considered mandatory in resource-limited settings [33], although WHO states that the combination of absence of any chest X-ray abnormality plus the absence of TB-related symptoms has the highest negative predictive value for ruling out TB [64]. Mobile vans equipped with a digital chest X-ray machine are increasingly being piloted and used in resource-limited settings. A study in Zimbabwe using a mobile van and digital chest X-ray showed that nearly 10% of asymptomatic persons with chest x-rays suggestive of pulmonary TB were diagnosed and treated for TB, with 13% of them found to have bacteriologically confirmed disease [65]. A similar study in India confirmed the value of chest X-ray in asymptomatic persons both in operational and economic terms [66]. Currently, digital chest X-rays are read by medical officers or other trained personnel. Accuracy of TB diagnosis can be improved using artificial intelligence to read the chest X-ray [67]: the automated technology is available and this should be considered and further researched and assessed by TB programmes where human resources are constrained. 

### 4.4. Expanding and Refining the Use of 3HP

A growing number of countries are using 3HP although there are several issues that require further research (Table 6). Caution is currently required before 3HP can be given safely to children < 2 years, pregnant women, injecting drug users on opioid substitution therapies (OST) and women using oral or injectable contraceptives. There is no published data on the use of 3HP in children < 2 years although a study is underway to assess safety and optimal dosing in this age group [43]. While 3HP was given to 125 pregnant women and showed rates of abortion and birth defects similar to those in the general population [68], this area needs further research in light of a randomised controlled trial on isoniazid preventive therapy showing a higher incidence of adverse pregnancy outcomes (stillbirth, low-birthweight, congenital anomalies) in HIV-infected women receiving isoniazid [69]. The risk of using 3HP in injecting drug users on Opioid Substitution Therapy (OST) is that the rifamycin component may lead to an “opiate withdrawal syndrome” due to decreased serum concentrations of the drugs [43]: this needs further study. 

All rifamycin-containing regimens have potential drug–drug interactions with ARV drugs, although in general rifapentine has less interaction than rifampicin. Dolutegravir (DTG) is now recommended as a preferred drug in first-line ARV regimens in PLHIV [70], and it has been established that 3HP can safely be used with this regimen without the need to adjust DTG doses [43]. 

In the 3HP regimen for adults, 900 mg rifapentine (6 × 150 mg tablets) is taken with 900 mg isoniazid (3 × 300 mg tablets) along with pyridoxine: 10 tablets on one day per week [43]. Lowering the pill burden (by offering rifapentine as a 300 mg tablet in a fixed-dose combination together with isoniazid 300 mg) would make 3HP more acceptable for people to take and would simplify procurement, distribution and storage issues at peripheral health facilities. These concerns are being taken up by generic drug manufacturers. 

Systematic monitoring is needed for common and important side effects. The most serious side effect of any isoniazid-containing regimen is drug-induced hepatitis [36], which if unrecognised can lead to acute liver failure and death. 3HP is associated with less hepatoxicity than 9-months IPT [71]. Nevertheless, programmes need to think about systematically excluding those at high risk of drug-induced hepatitis (for example, with pre-existing liver disease or chronic hepatitis C infection) and monitor this aspect closely. Given the absence of laboratory monitoring in most resource-constrained countries, those taking 3HP and health workers must be educated about the symptoms and signs of hepatitis and the need to stop the drug and immediately report to a health facility if these occur. 

3HP is said to be more cost-effective when given by clinic-based direct observation (DOT) [72]. However, in the USA, self-administered 3HP with monthly monitoring with or without weekly text messaging was non-inferior to 3HP by DOT in terms of safety and treatment completion [73]. Video observed therapy (VOT) has also emerged recently as a method to mimic in-person visits, especially in the smartphone era with internet data connections. VOT was associated with higher treatment completion in persons taking 3HP compared with DOT in New York [74], and in South India VOT was preferred over DOT in terms of support during care and treatment of TB [75]. This is an important research topic in low- and middle-income countries where local information on demographics and smartphone ownership is crucial to understand who might and might not benefit from this digital technology.

TB preventive therapy reduces but does not completely prevent TB, and all individuals must be monitored for the development of active TB during treatment, and, if possible, after treatment as well. This requires education of those initiated on preventive therapy as well as their attending health care workers, with clear instructions to attend health facilities for screening and investigation if suggestive TB symptoms arise. For PLHIV living in high TB exposure environments, there is a need to determine whether repeat courses of 3HP are required to maintain TB preventive effects. 

### 4.5. Recording and Reporting

Keeping track of who is eligible, who initiates, who completes TB preventive therapy and who is free of TB 12 months after completing therapy is essential for (i) monitoring each individual’s journey, (ii) assessing the TB preventive therapy cascade, (iii) charting the progress made against indicators (such as rates of coverage, completion or failure) at subnational, national and international level and (iv) drug forecasting so that procurement and distribution match demand. Drug shortages and interrupted supplies were the most common reasons for discontinuing IPT in children in an Ethiopian community-based LTBI treatment study [76]. 

In PLHIV, National HIV/AIDS Programmes take responsibility for recording and reporting on who is screened for TB, diagnosed with TB and given TB preventive therapy. These data, collated annually for countries and at global level, are usually presented in the Global TB Reports [4]. 

For household contacts of index TB patients and for all other high-risk groups, the National TB Programme generally takes responsibility. Done properly, this is an enormous task requiring adequate human, financial and technical resources. To fully comprehend how all the steps of preventive therapy work at the programme level, a sufficient amount of detail must initially be collected. Table 7 outlines the key indicators for which data should be collected in household contacts of index patients with TB. This should identify bottlenecks or problem areas where operational research or further work might be required to close gaps in the TB preventive therapy cascade and better streamline activities. If the index patient is HIV-positive or has DM, further testing of household contacts with respect to these parameters would be indicated.

### 4.6. Consideration of Other TB Prevention Activities

TB preventive therapy is an important part of a larger effort to prevent TB. The development and widespread use of an effective vaccine would have an enormous impact. BCG vaccine protects children from severe disease such as disseminated TB and meningitis, but it does not afford long term protection against pulmonary disease. However, a novel candidate vaccine, M72/AS01_E_, provided 50% protection over three years against progression to pulmonary TB in adults with LTBI enrolled in Kenya, South Africa and Zambia [77]. While more work on this vaccine is needed in different populations and age-groups as well as people with no evidence of LTBI, this is an exciting and promising development. 

Intervening on socioeconomic and other determinants of TB can yield valuable preventive dividends. For example, ART is an excellent TB prevention tool in PLHIV [78], and in South Korea the use of metformin in elderly people with DM significantly reduced their risk of TB [79]. On a much larger scale, poverty reduction at the country level predictably reduces TB incidence [80], and targeted socioeconomic poverty reduction interventions such as cash transfers can also reduce TB risk [81]. Good infection control policies and practices in health care facilities and congregate settings such as refugee camps and prisons can reduce *MTB* transmission and lower TB incidence. 

## 5. Conclusions

The Lancet Commission on TB outlined two main strategies for better progress towards ending the TB epidemic [82], and they both apply to TB preventive therapy. First, we need to improve the implementation of proven interventions and ensure efficient and rapid scale-up as described earlier in this perspective. This applies to finding high risk-groups, considering the expanded use of chest X-ray, properly assessing who is eligible for preventive therapy, initiating appropriate and acceptable treatment and ensuring that a course of treatment is completed with due attention to safety and medication adherence. It is also important to see how TB preventive services can be integrated with those that are already established for diagnosis and treatment. Second, we need to invest in and deploy new products, the most important of which would be an affordable, reliable and easy-to-use biomarker to predict who is at risk of progressing to active disease. More effort is also needed to embrace digital technology, not only for diagnostic tools and monitoring treatment adherence, but for data services, recording and reporting and health service management [83]. 

For years, the diagnosis and treatment of TB has been the cornerstone of TB control efforts. Given the inefficiencies of this process, the ensuing morbidity and mortality that accompany the treatment period and the recognition of disability and enhanced mortality after treatment is completed, prevention has to be better embraced, properly implemented and scaled up. As Benjamin Franklin famously stated almost 300 years ago “an ounce of prevention is worth a pound of cure”. 

## Figures and Tables

**Table 1 tropicalmed-05-00061-t001:** Tuberculosis (TB) preventive treatment in 2018.

WHO Region	PLHIV Newly Enrolled in Care Who Were Given TB Preventive Treatment ^a^ %	Household Children (aged < 5) Contacts of Bacteriologically Confirmed TB Patients Who Were Given TB Preventive Treatment %
Africa	60	29
Americas	9	55
Eastern Mediterranean	13	23
European	69	>100
South-East Asia	15	26
Western Pacific	39	12
Global	49	27

WHO = World Health Organization; PLHIV = persons living with HIV. ^a^ Calculations exclude countries with missing numerators or denominators. Asia Pacific comprises South-East Asia and the Western Pacific Adapted from [4].

**Table 2 tropicalmed-05-00061-t002:** Treatment success in cohorts of TB patients registered for treatment.

**2A: Global**		
Cohorts of TB patients registered for treatment	Registered in cohort N	Treatment Success %
New/previously treated patients registered in 2017 ^a^	6,381,295	84
HIV-positive TB patients registered in 2017	445,922	75
MDR/RR-TB patients started on SLD in 2016	126,089	56
XDR-TB patients started on SLD in 2016	9258	39
**2B: South-East Asia Region**		
Cohorts of TB patients registered for treatment	Registered in cohort N	Treatment Success %
New/previously treated patients registered in 2017 ^a^	2,746,023	82
HIV-positive TB patients registered in 2017	56,872	71
MDR/RR-TB patients started on SLD in 2016	40,725	52
XDR-TB patients started on SLD in 2016	2567	31
**2C: Western Pacific Region**		
Cohorts of TB patients registered for treatment	Registered in cohort N	Treatment Success %
New/previously treated patients registered in 2017 ^a^	1,360,505	91
HIV-positive TB patients registered in 2017	12,170	79
MDR/RR-TB patients started on SLD in 2016	14,602	59
XDR-TB patients started on SLD in 2016	88	58

^a^ Some countries reported on new patients only. Treatment success = cured and treatment completed; MDR-TB = multidrug-resistant TB; RR-TB = rifampicin-resistant TB; XDR-TB = extensively drug-resistant TB; SLD = second-line anti-TB drugs. Asia Pacific comprises South-East Asia and the Western Pacific. Adapted from [4].

**Table 3 tropicalmed-05-00061-t003:** Testing and treatment of latent TB infection (LTBI) in HIV-negative high-risk groups who are not household contacts.

Category	Type of Person	Need for Systematic Testing of LTBI	Treatment of LTBI
1	Patient with Silicosis	Yes	Recommended if LTBI test is positive
2	Patients in end-stage renal failure receiving dialysis Patients who have received haematological or organ transplants Patients receiving tumour necrosis factor α-neutralising agents for Crohn’s disease or rheumatoid arthritis Patients using oral or inhaled corticosteroids	Yes for all Category 2	Recommended if LTBI test is positive for all Category 2
3	In countries with low TB incidence: Prisoners Homeless people People who inject drugs Health care workers Immigrants from countries with a high TB burden	Yes for all Category 3	Recommended if LTBI test is positive for all Category 3
4	Persons with diabetes mellitus Persons with harmful alcohol consumption People who smoke tobacco People who are underweight	No for all Category 4	Recommended if LTBI testing is done on individual basis and LTBI test is found to be positive

TB = tuberculosis; LTBI = latent tuberculosis infection. Adapted from [33] and [36].

**Table 4 tropicalmed-05-00061-t004:** Alternative shorter TB preventive therapy regimens.

Treatment Regimen	Duration	Dosage Frequency	Common Abbreviation
Rifampicin	3–4 months	Daily	3R/4R
Rifampicin and isoniazid	3–4 months	Daily	3RH/4RH
Rifapentine and isoniazid	3 months	Weekly	3HP
Rifapentine and isoniazid	4 weeks	Daily	1HP

**Table 5 tropicalmed-05-00061-t005:** Research and innovation on the index patient and their household contacts.

Research questions around index patient	Determine the value in assessing the index patient’s drug susceptibility status in order to better prescribe the type of TB preventive therapy for household contacts
	Assess whether the index patient should be systematically screened for risk factors such as HIV, DM, smoking, alcohol abuse and malnutrition
Research questions around household contacts	Clarify the definition of household contact for the local context
	Assess whether household contact screening should be done just for index patients with bacteriologically confirmed pulmonary TB or include index patients with clinically diagnosed pulmonary TB
	Explore whether household contacts should be systematically screened for risk factors such as HIV, DM, smoking, alcohol abuse and malnutrition irrespective of whether the index patient has these risk factors
	In countries that still insist on LTBI testing of household contacts, assess in the local context whether this is needed or whether all household contacts can just be treated

TB = tuberculosis; HIV = human immunodeficiency virus; DM = diabetes mellitus.

**Table 6 tropicalmed-05-00061-t006:** Research and innovation on the 3-month weekly rifapentine and isoniazid (3HP) treatment regimen.

Issues Around 3HP	Category	Research and Evidence Needed
Caution and safety	Children < 2 years	Acceptability of water-dispersible formulations: one trial underway
	Pregnant women	Frequency of maternal adverse events and pregnancy adverse outcomes
	PWID on OST	Frequency of opiate withdrawal syndrome and measures needed to avoid it
	Women on oral or injectable contraceptives	Interactions with contraceptives and possible dosage adjustments
	Drug-drug interactions in PLHIV	3HP interactions with nevirapine, efavirenz and protease inhibitors
Acceptable formulations	Pill burden: 10 pills once a week: 6 pills of rifapentine 3 pills of isoniazid 1 pill of pyridoxine	Simpler fixed-dose combination—e.g., three tablets combined rifapentine (300 mg) and isoniazid (300 mg) once a week
Monitoring for adverse events	Drug-induced hepatitis and acute liver failure	How to monitor without laboratory infrastructure and how to educate people and health care workers about hepatitis and acute liver failure
Administration of medication	Clinic-based DOT or self-administered treatment or VOT through smartphones	Locally based operational research on how best to administer 3HP in terms of medication adherence, safety and treatment completion
Number of courses of 3HP	PLHIV living in high TB exposure environments	The need, if any, of repeat courses of 3HP to further reduce risk of TB and the frequency of these repeat courses

3HP = 3 months of weekly isoniazid and rifapentine; PWID = people who inject drugs: OST = opioid substitution therapies; PLHIV = people living with HIV; DOT = directly observed therapy; VOT = video-observed therapy.

**Table 7 tropicalmed-05-00061-t007:** TB Preventive therapy master card for household contacts of index patients with TB. Index Patient Details: Name; registration number: type and category of TB; age; sex; cigarette smoker; consumes alcohol; HIV status; diabetes mellitus status. Line List of Household contacts with details of TPT.

Name	Age	Sex	Relationship to index patient	Symptom Screen Positive Negative Not done	CXR Positive Negative Not done	Active TB diagnosis Yes No	Eligible for TPT Yes No	Reason for non-eligibility ^a^	**TPT started** Yes (Date) No	**TPT completed** Yes (Date) No	**TB status at 12 months after TPT completion** No TB (Date) TB (Date)

^a^ Reasons = known alcohol abuse; acute hepatitis; chronic liver disease; infection with hepatitis B or C; other TB = tuberculosis; CXR = chest X-ray; TPT = TB preventive therapy; Note: if index patient is HIV-positive then screen household contacts for HIV: stratify the numbers below for HIV.
Number of household contacts < 5 years: Number of other household contacts:Number diagnosed with TB:Number diagnosed with TB:Number given TPT:Number given TPT:Number finished TPT:Number finished TPT:

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
