# Peer review of "The Growing Importance of Tuberculosis Preventive Therapy and How Research and Innovation Can Enhance Its Implementation on the Ground"

_tropicalmed, 2020, doi:10.3390/tropicalmed5020061_

Round 1
Reviewer 1 Report
This manuscript has discussed the current issues about TB epidemiology, diagnosis and treatment. The references are up to date. Although this review has not covered newer topics like the screening of patients before monoclonal therapy (e.g for RA or cancer therapy) as well as the role of host-directed therapy in managing LTBI, authors stated in detail the other issues comprehensively. As a minor comment, the format the tables were made benefits from some clearer format.
Reviewer 2 Report
In this perspective Harries and colleagues discuss several aspects of tuberculosis preventative therapy and how it can be implemented more effectively around the globe.
The authors first outline the importance of TB management and why the prevention of TB is an under-utilized aspect of TB control. They go on to discuss the challenges currently preventing more widespread use of preventative therapy. The authors discuss the challenges and problems around diagnosis of active TB vs. LTBI and how it influences the decision and ability of different TB control programs to initiate and monitor preventive therapy.
They go on to outline the current status and management of TB preventative therapy and why it is important to identify at-risk groups. They give a comprehensive summary of currently used treatment regimens and how these could be improved with newer drug combinations.
Most importantly, the authors outline how research and innovation can and should improve the uptake of TB preventative therapy. In particular the issue of better diagnostic tests to determine the difference between true latent infection and immunological memory of a past infection is discussed. How do you rule out active TB and how can IGRAs be improved? All very relevant questions for the management of LTBI. The authors make recommendations on how recording and reporting of TB preventative therapy could be improved, and how bottlenecks around human resources and logistics could be overcome. They provide a template for a recording card that could be used to collect appropriate indicators for each index and contact.
Unfortunately, the section on one of the most effective preventative measures - vaccination - falls short of its full potential and should be expanded. The authors mention the promising M72 results, but do not go into any detail regarding other vaccine candidates (VPM, MTBVAC etc) and the recent promising BCG re-vaccination results. Also, the positive impact on TB prevention due to socio-economic interventions are only touched upon in a couple of sentences. These sections should be expanded.
Furthermore, the style of all Tables is odd (perhaps this is due to the PDF version I have been provided with or a journal requirement). The tables could be made more user-friendly and appealing.
Overall, the article is well-written, well referenced and topical. The authors are experts in the field and respected authorities in their respective areas.
Reviewer 3 Report
General comments
This manuscript describes a transition in recommendations for increasing the frequency of diagnosis and treatment of latent TB infection (LTBI) from very limited WHO criteria to much broader ones in order to accelerate progress towards the goal of TB elimination. The list of risk-factors for progression from LTBI recommended to identify individuals at a higher priority for treatment (Table 3) goes beyond the prior WHO recommendations to treat only young household contacts and PLHIV (even without LTBI testing), includes testing with either TST or IGRA and recommends new, short-course regimens.
There is one population at risk for progression to active TB that was not mentioned in the WHO guideline or in this manuscript: individuals with chest radiographic findings of fibrotic lung lesions consistent with inactive TB, a population among whom the IUAT trial of INH for 12, 24 and 48 weeks was compared to placebo in RCT. Perhaps this population was not addressed by WHO due to the limited access to quality chest radiography in many high-burden countries. This limitation can also be surmounted with mobile units and potentially with artificial intelligence support as discussed in this manuscript.
I believe it is important to address this issue in this manuscript for two reasons. First, the publication by Onozaki et al in Tropical Medicine and Public Health 2015 noted that 40-79% of individuals in Asia with culture confirmed TB diagnosed during national prevalence surveys were asymptomatic. TB case-finding based on symptoms is clearly inadequate and adds to the gaps in treatment initiation and completion. A lack of symptoms is also inadequate for excluding TB in individuals with a positive LTBI test. Second, the population with chest radiographic findings but negative cultures in these surveys represent culture-negative TB to some extent, but individuals with LTBI and “fibrotic lung lesions” after active TB is excluded represent a group with a 5-fold risk of TB. Expanding LTBI testing thus will also be a TB case-finding activity in high TB-risk populations. There is no doubt about the challenges of accomplishing expanded chest radiography, but without the as-yet-discovered ideal test to distinguish TB from LTB there seems to be little alternative.
There is considerable evidence of an impact from detecting and treating these individuals with asymptomatic, minimal-lesion, usually AFB smear-negative TB. A number of studies of immigrant applicants for U.S. permanent residency who are screened pre-departure and followed up post-arrival provide such data. The TB rates drop from 297 in the first year after arrival to 42.5 per 100k for years 2-10 among new immigrants from the Philippines (C. Tsang EID 2020). Almost all of this drop is likely due to prevalent TB case detection by chest radiography and culture followed by treatment pre-departure. In another study, overall 14% of Filipino adult applicants had abnormal chest radiographs and 1% of adults had culture-positive TB pre-departure. Annual TB rates post-arrival were stable for 9 years at 32 for those with normal and 3-fold higher for those with abnormal chest X-rays. An unknown proportion of the latter were treated for LTBI but probably few of the others. (N Walter AJRCCM 2013).
Specific comments
Abstract – If information about the potential and benefit of expanding chest radiography, some revision may be needed.
Background – Please include the information about the prevalence of culture-positive TB in Asia among asymptomatic adults and means of diagnosis with implications for expanding LTBI diagnosis and treatment.
Table 5 – Is there doubt about the need to screen patients with active TB for those factors?
P 10, para line 298: Consider expanding upon the benefits of chest radiography
Discussion - Some revision may be indicated if expanded radiography is discussed.
